# Prevalence, Persistence, and Agreement of Physical Frailty Tools in Patients with Severe COPD Declining Pulmonary Rehabilitation: An Exploratory 1-Year Prospective Cohort Study [note 1]

**DOI:** 10.3390/jcm14186434

**Published:** 2025-09-12

**Authors:** Henrik Hansen, Jeanette Hansen, Christina Nielsen, Nina Godtfredsen

**Affiliations:** 1Respiratory Research Unit, Department of Respiratory Medicine, Copenhagen University Hospital, 2650 Hvidovre, Denmark; christina.nielsen.07@regionh.dk (C.N.); nina.skavlan.godtfredsen@regionh.dk (N.G.); 2Institute of Rehabilitation Sciences, University of Antwerp, 2000 Antwerpen, Belgium; 3Institute of Clinical Medicine, University of Copenhagen, 2100 Copenhagen, Denmark; 4Department of Midwifery, Physiotherapy, Occupational Therapy and Psychomotor Therapy, Faculty of Health, University College Copenhagen, 2200 Copenhagen, Denmark; jeah@kp.dk

**Keywords:** COPD, physical frailty, prospective cohort study, extrapulmonary traits, personalized medicine

## Abstract

**Background:** Physical frailty is a prevalent and clinically important manifestation of COPD. While the ERS/ATS recommends the Short Physical Performance Battery (SPPB), handgrip strength (HGS), 30 s sit-to-stand (30secSTS), and Timed-Up-and-Go (TUG) as frailty screening tools, their agreement and predictive performance remain unclear. Furthermore, the trajectory of frailty is poorly understood in patients who decline pulmonary rehabilitation (PR). **Objective:** To assess the prevalence of and change in physical frailty and its association with 12-month all-cause hospitalizations and mortality. Secondarily, to assess the agreement and predictive value (positive; PPV/negative; NPV) of recommended screening tests in COPD patients declining PR. **Methods:** In this prospective cohort study, 102 patients with COPD (61 females, mean ± SD age 70 ± 9 years, FEV_1_ 34 ± 11%, SPPB 8.0 ± 3.2 points, CAT 19 ± 7) underwent repeated frailty assessments at baseline and after 12 months using the SPPB (reference), TUG, 30secSTS, and HGS. **Results:** At baseline, 39% were physically frail (SPPB ≤ 7). Frailty persisted in 86%, and 23% had died at 12 months. Baseline age-adjusted physical frailty was not statistically associated with 12-month all-cause hospitalization (OR 1.79 [0.61–5.24]) or mortality (OR 3.54 [0.95–13.16]). Agreement with SPPB was moderate for TUG (κ = 0.53) and fair for 30secSTS (κ = 0.38) and HGS (κ = 0.26), with similar findings at 12 months. TUG had the highest PPV/NPV (0.85/0.71). **Conclusions:** Physical frailty is prevalent and persistent in patients with severe COPD who decline PR. ERS/ATS-recommended tools showed fair to moderate agreement and predictive value. TUG was the most robust proxy, though tool selection should be guided by clinical context and purpose.

## 1. Introduction

An increasing number of people are living with frailty [1]. Frailty is characterized by a decline in functioning across multiple physiological systems, accompanied by an increased vulnerability and physical disability [2,3]. Having frailty places a person at increased risk of adverse outcomes, including falls, hospitalization, and mortality [2]. Studies have shown a clear pattern of increased health-care use and costs associated with frailty [2]. All older adults are at risk of developing frailty, although risk levels are substantially higher among those with multiple comorbidities, low socioeconomic position, poor diet, and sedentary lifestyle [1,2]. The concept of frailty is increasingly being used in primary, acute, and specialist healthcare. However, despite efforts over the past three decades, agreement on a standard instrument to identify frailty is only lately emerging [1,2,4,5,6].

A customary measure of frailty is the Fried Frailty Phenotype (FFP), which consists of five criteria: unintentional weight loss, weakness or poor handgrip strength, self-reported exhaustion, slow walking speed, and low physical activity [3,7]. Additionally, the Short Physical Performance Battery (SPPB) is recommended for describing physical frailty in people aged ≥ 65 years by combining the results of static balance tests, gait speed (GS), and the five-times sit-to-stand test [6,8,9]. Based on previous studies on risk of disability and mortality, summary cut-off scores for the SPPB have been defined as follows: 10–12 points = normal, 8–9 points = pre-frail patient, and ≤7 points = frail patient [6,10,11,12]. The SPPB has also demonstrated utility in predicting adverse events, functional decline, mortality, and change following pulmonary rehabilitation [13,14]. A study by Zhang et al. demonstrated that SPPB was non-inferior to FFP in predicting 1-year mortality in older patients with stable COPD [15].

Hand grip strength (HGS) is a key component in several frailty models and is independently a strong predictor for readmission and mortality risk [16,17]. HGS is also included in the European algorithm for assessing sarcopenia, with cut-off scores of 27 kg for men and 16 kg for women, although these cut-off scores apply only to older people (≥65 years) [5]. Based on three Danish population surveys (n = 29.614) and additional Danish reference material [18,19,20,21], new Danish reference values have been developed for every age decade from 18 to 90+ years [22]. Correspondingly, Danish reference values have been developed for the 30 s sit-to-stand test (30secSTS), providing age-specific norms across adulthood [22].

The 30secSTS and the Timed-Up-and-Go test (TUG) are two increasingly used assessment tests validated for screening for physically frail individuals [6,23,24]. Both tests are strongly associated with fall risk, physical frailty, and mortality [23,24,25]. While normative values exist for the 30secSTS in the general population, a specific cut-off of >8 s has been proposed for the TUG test in patients with COPD, as outlined in the recent frailty guidelines from the American Thoracic Society (ATS) [6,26]. To date, the agreement between the defined cut-offs for physical frailty across these four tests, as well as their stability over time, has not been systematically investigated in patients with COPD or any other respiratory disease.

People with COPD are recognized to have a higher prevalence of sarcopenia and frailty than the general population [4,6,27,28]. However, the prevalence is likely to differ by different frailty definitions and assessment tools, as well as in different clinical settings and disease severity stages [4,27]. First-line management of physical frailty in people with COPD includes a multi-component rehabilitation program with emphasis on resistance-, balance-, and functional-based training, and it has proven effective in outpatient pulmonary rehabilitation programs [6,29,30]. While prevalence of frailty and predictive models for hospitalization and mortality have been investigated for the past decade [31,32], there remains limited evidence on the persistency and transitions of frailty over time, particularly among patients with COPD who decline pulmonary rehabilitation—a group most vulnerable to poor outcomes and least studied in this context. Moreover, the agreement and predictive accuracy of ERS/ATS-recommended screening tools [4,6] in this population has not been systematically investigated.

Thus, this exploratory study aimed to investigate (a) the prevalence and 12-month stability of physical frailty in patients with severe COPD who declined participation in PR; (b) the association between baseline physical frailty and 12-month all-cause hospitalization and mortality; and (c) the evaluation of kappa agreement, positive predictive value (PPV), and negative predictive value (NPV) of the four ERS/ATS-recommended physical frailty screening tests [4,6], and their 12-month stability.

## 2. Methods

### 2.1. Study Design and Participants

This exploratory prospective 12-month cohort study was based on data from an existing clinical cohort study (NCT04249388). Patients were consecutively included from four respiratory departments across hospitals in Greater Copenhagen during routine consultations from June 2020 to December 2021 at the time they declined participation in an outpatient hospital-based routine PR. Recruitment for PR was possible from June 2020, following the reopening of society after the first COVID-19 lockdown; therefore, outpatient PR was offered at routine control visits. None of the patients recruited for this study reported COVID-19 as a reason for declining outpatient hospital-based routine PR. The Danish Data Protection Agency approved the research database (P-2019-730). As this study was noninterventional, ethical approval was not required according to Danish law. Informed verbal and written consent was obtained from all subjects involved in the study.

The cohort consisted of patients with physician-diagnosed COPD (FEV_1_/FVC < 0.7), FEV_1_ < 50% predicted (spirometry GOLD stages 3–4), and a Medical Research Council (MRC) dyspnea scale ≥ 2, who declined to participate in an outpatient hospital-based PR program. Exclusion criteria included completion of PR within the previous twelve months, dementia or cognitive impairment, uncontrolled psychiatric illness, language difficulties that prevented understanding instructions or completing patient-reported outcome measures, and comorbidities contraindicated to the assessment protocol. The study reports according to the STROBE guideline for cohort studies [33].

### 2.2. Assessments

All assessors completed a two-hour assessment course to ensure consistent and standardized protocol procedures, including programming of sensors (ActivePAL triaxial accelerometer, PAL Technologies Ltd., Glasgow, UK)) and recording of results. In addition, assessors observed a minimum of four live assessments before being accredited. At the 12-month follow-up, both assessors and patients were blinded to baseline outcomes.

The assessment procedure reflected everyday clinical practice, where multiple performance tests and questionnaires were conducted within a limited time frame. The procedures are reproducible, and detailed descriptions have been published previously [34,35].

### 2.3. Variables

#### Short Physical Performance Battery (SPPB)

The primary variable for categorizing patients’ physical status was the Short Physical Performance Battery (SPPB). The SPPB scores were classified into physical frail (≤7), physical pre-frail (8–9), and normal (≥10) according to European Respiratory Society (ERS) and American Thoracic Society (ATS) guidelines on frailty in adults with chronic respiratory diseases [4,6].

The SPPB consists of a standing balance test, 3 or 4 m gait speed (3/4MGS), and the 5-times-sit-to-stand (5STS) test, conducted according to the National Institute on Aging protocol [10]. The balance test requires participants to hold three progressively difficult stances (side-by-side, semi-tandem, and tandem) for 10 s each. The 3MGS measures the time to walk 3 m at a habitual pace, with the faster of two trials recorded in seconds. In the 5STS, participants stand up from a chair five times with arms across the chest, with the faster of two trials recorded in seconds. Each component is scored from 0 to 4, yielding a total SPPB score ranging from 0 (functional impairment) to 12 (maximal capacity).

### 2.4. Handgrip Strength (HGS)

Handgrip strength (HGS) was assessed using a hand-held dynamometer (Jamar dynamometer). The HGS was measured with the dominant and non-dominant hand in a seated position. The arm was kept slightly abducted, the elbow flexed at 90 degrees, and the forearm in a neutral position. Three attempts were performed with the dominant and non-dominant side, respectively. The highest value obtained from the dominant side was selected for statistical analysis and cut-off categorization. Cut-off values for low HGS were based on age- and sex-adjusted Danish national reference values, which are categorized as normal, below one standard deviation (SD), and below two SD [22]. We collapsed below one and two SD as “reduced”, corresponding to values below the 16th percentile.

### 2.5. 30 s Sit-to-Stand Test (30secSTS)

The same chair, with a seat height of 45–47 cm, was used throughout all tests, and the patients were instructed to stand up fully and sit down as many times as possible in 30 s with arms across the chest, starting from the seated position. The number of full stands was recorded as the total number of repetitions. A score of zero was recorded if the patient was unable to rise from the chair without using the arms. A 30 min pause between the first and second trials was mandatory in this study, with the higher number of repetitions of the two trials recorded. Cut-off values for low 30secSTS were based on age- and sex-adjusted Danish national reference values, which are categorized as normal, below one standard deviation (SD), and below two SD [22]. We collapsed below one and two SD as “reduced”, corresponding to values below the 16th percentile.

### 2.6. Timed-Up-and-Go (TUG)

All participants performed the TUG according to published protocol instructions [36]. Participants were instructed to stand up from a standardized chair (seat height 45–47 cm), walk at a comfortable and safe pace for three meters to a mark on the floor, turn, and walk back to the chair to sit down [36]. The fastest of three trials was recorded in seconds. Participants were allowed to rest for one minute between the trials. We determined the cut-off values for abnormal TUG as ≥8 s as proposed for patients with COPD in the recent 2023 ATS frailty guideline report [6].

Participants on long-term oxygen treatment (LTOT) were asked to carry the ambulatory oxygen device during all performance tests.

### 2.7. 12-Month All-Cause Hospitalization and All-Cause Mortality

The total number of all-cause hospitalizations and deaths from baseline (T0) to 12-month follow-up (T1) was registered from the hospital records.

### 2.8. Descriptive Variables

Baseline descriptive variables included sex, age, marital status, body mass index (BMI), forced expiratory volume in one second (FEV_1_), long-term oxygen treatment (LTOT), smoking status, Charlson Comorbidity Index (CCI), number of steps per day, use of a walking aid, and previous experience with pulmonary rehabilitation. Patient-reported outcomes were also collected: fatigue (Multidimensional Fatigue Inventory, MFI-20), pain (Brief Pain Inventory, BPI), health-related quality of life (EQ-5D utility index and visual analog scale), and symptoms of anxiety and depression (Hospital Anxiety and Depression Scale, HADS). Data on years of education and inhaler medication were collected but not reported.

### 2.9. Sample Size Considerations

The sample size for this cohort was originally determined for the primary aim of the parent study (ClinicalTrials.gov ID: NCT04249388), which compared characteristics of patients with severe COPD opting in or out of PR. The sample size was calculated for a cross-sectional comparison of SPPB scores between two groups, assuming a standard deviation of 5.0 points, a 1-point precision, and a 5% Type I error. The present study constitutes a secondary prospective analysis and is therefore considered exploratory.

#### Statistics

Baseline differences across SPPB-classified physical frailty groups (normal, pre-frail, and frail) were examined using chi square tests for categorical variables and one-way ANOVA for continuous variables with normally distributed data. For non-normally distributed variables, the Kruskal–Wallis test was applied. Prevalence is presented as percentages. Changes in frailty classification from baseline (T0) to 12 months (T1) were visualized using Sankey plots and assessed using Bowker’s test of symmetry, appropriate for paired categorical data with more than two levels. The changes in test scores from T0 to T1 were tested with a paired *t*-test and Wilcoxon signed-rank test depending on distribution and presented as mean difference (SE 95% CI) or median difference (25th to 75th IQR).

The association between baseline frailty status and 12-month all-cause mortality and all-cause hospitalization (≥1) was investigated with unadjusted and age-adjusted binomial logistic regression analyses and presented as odds ratios with 95% confidence intervals. Due to the small sample size, and for a conservative estimation, the normal and pre-frail groups were combined and used as the reference category in these models.

Agreement between SPPB-based frailty classification and the classifications of frailty from HGS, 30secSTS, and TUG was tested with unweighted kappa and interpreted according to Landis and Koch (1977) as follows: <0.20 = slight, 0.21–0.40 = fair, 0.41–0.60 = moderate, 0.61–0.80 = substantial, and >0.80 = almost perfect agreement [37]. Positive (PPV = true positive/true positive + false positive) and negative (NPV = true negative/true negative + false negative) predictive values were also calculated with SPPB classification as the benchmark. Categorizations were dichotomized for simplification, and a cut-off of <10 points (pre-frail + frail) on the SPPB to define those at risk was selected based on clinical relevance to identify early functional decline rather than only capturing established physical frailty. Details on the dichotomization of HGS, 30secSTS, and TUG are provided in the variables section. Predictive values were interpreted according to Fleiss 2013 as follows: <0.60 = low, 0.60–0.79 = moderate, and >0.80 = high [38].

## 3. Results

Of 164 patients initially declining PR and meeting the inclusion criteria, 102 accepted to participate in the study. The 62 patients not included did not differ in sex, age, or FEV_1_ (45% female, mean ± SD age 70.6 ± 10.0 years, FEV_1_ 33.2 ± 11.1%) compared to the included patients (Table 1).

Baseline characteristics of participants are shown in Table 1. The participants had severe airflow obstruction, a high burden of respiratory symptoms, and a moderate–high Charlson comorbidity index score. The median time since COPD diagnosis was 10 years (IQR: 6–20), and the median time since previous PR participation was 5 years (IQR: 3; 6). Thirty-eight percent were naïve to PR. At baseline, patients classified as physically pre-frail and frail were older, more often female, had lower physical performance (SPPB, HGS, 30secSTS, TUG), fewer daily steps, poorer self-reported health (EQ-5D VAS), and higher fatigue scores (Table 1, *p* < 0.05).

The prevalence of physical frailty at baseline (T0) was 39%, as indicated by a Short Physical Performance Battery (SPPB) score of ≤7 points. Among patients with complete datasets at the 12-month follow-up (T1), the prevalence of physical frailty was 36%. For the entire cohort, physical frailty prevalence decreased to 25% at T1. This change was due to nine patients who were frail at baseline having died before T1, six patients who had non-ambulatory functioning, one patient who was in palliative care hospice, and three patients who declined follow-up assessment, preventing classification by the SPPB (Figure 1—Sankey plot).

Of those classified as physically frail at T0 with complete data, 86% (18/21) remained physically frail, 14% (3/21) improved to pre-frail, and none was classified as normal at T1. For the pre-frail group, 44% (8/18) remained pre-frail, 33% (6/18) became physically frail, and 23% (4/18) improved to normal. Among those classified as physically normal at T0, 81% remained normal, 16% regressed to pre-frail, and 3% became frail at T1. Bowker’s test for symmetry showed no significant asymmetry in transitions between frailty states (χ^2^(df = 3) = 2.11, *p* = 0.550), suggesting no significant difference in transition proportions between categories.

The changes in test score from T0 to T1 were statistically significantly different for the 30secSTS for the whole group and within the baseline frail classification and similar for TUG in the whole group (Table 2). None of the changes exceeded the established minimal clinically important difference.

In the age-adjusted binomial regression analysis, baseline frailty was not statistically associated with all-cause hospitalization (OR 1.79 [0.61; 5.24]) or all-cause mortality at 12-month follow-up (OR 3.54 [0.95; 13.16]).

Using SPPB (<10 points indicating frail or pre-frail) as the reference for physical frailty and applying guideline-based cut-offs for TUG, 30secSTS, and HGS, both kappa agreement and predictive values remained stable over 12 months (Table 3). TUG showed moderate agreement (κ = 0.53) at both time points, with consistently moderate-to-high PPV and NPV. The 30secSTS demonstrated fair agreement (κ = 0.38) and similarly favorable predictive values. HGS, by contrast, showed only fair agreement (κ = 0.26), with moderate PPVs but persistently low NPVs (Table 3).

## 4. Discussion

This study found that physical frailty was present in approximately 36–39% of patients with severe COPD who declined participation in PR and that frailty status remained largely unchanged over 12 months. Most patients classified as frail or pre-frail at baseline either maintained or worsened their status in the absence of structured PR or other exercise interventions. Baseline frailty, adjusted for age, was not significantly associated with all-cause hospitalization or mortality at 12 months. Agreement, PPV, and NPV of the four ERS/ATS-recommended screening tools for physical frailty were fair to moderate and remained stable over 12 months.

Although our cohort represents a specific severity subgroup opting for PR, the observed prevalence of physical frailty aligns with previous studies reporting a pooled mean prevalence of 32.07% (95% confidence interval (CI) 26.64–37.49) [31] in COPD populations, depending on frailty criteria applied, COPD severity, and clinical setting (e.g., nursing home, community, hospital) [1,30,31,32].

To our knowledge, this is only the second study [39] to report frailty transitions over time in patients with COPD and the first to focus specifically on individuals abstaining from structured exercise or rehabilitation. In contrast to our findings, a 2-year cohort study by Bernabeu-Mora et al. (2020) reported a lower frailty prevalence (8%) and a higher rate of improvement in frailty status (18% vs. our 7%) [39]. Several factors likely explain these differences. While both cohorts were recruited from outpatient respiratory clinics, the Spanish cohort included more men (87% vs. 50%), had higher baseline SPPB scores (9.7 vs. 8.0), had milder COPD (FEV_1_% 50 vs. 34; CAT score 13 vs. 19), and applied the Fried Frailty Phenotype rather than the SPPB. Moreover, their analysis did not account for participation in rehabilitation programs. Thus, the studies offer complementary, rather than directly comparable, insight into frailty transition over time. Unlike Bernabeu-Mora et al. [39], we statistically assessed frailty transitions and found no significant change in frailty or pre-frailty status over 12 months. This highlights a critical clinical insight: in the absence of structured exercise, frailty and functional limitations are unlikely to improve (Figure 1, Table 2). Importantly, existing evidence suggests that frailty is at least partially reversible with pulmonary rehabilitation [30,31,40].

Contrary to previous studies [1,2,4,5,6,31,32], we did not find a statistically significant association between baseline frailty and all-cause hospitalization or mortality at 12 months. While point estimates were directionally consistent with earlier findings, wide confidence intervals suggest that limited statistical power (n = 102) may account for the lack of significance.

While previous respiratory and geriatric studies have assessed individual frailty tools, none, to our knowledge, have directly compared the performance of HGS, 30secSTS, and TUG [5,6,22] against the SPPB using clinically interpretable agreement metrics and predictive accuracy in COPD [4]. The prior literature supports the construct validity of these tools, though typically through correlation, regression, or AUC analyses [6,15,41,42]. Our findings provide clinically relevant, head-to-head comparisons using kappa agreement, PPV, and NPV, which may better inform tool selection for initial frailty screening and longitudinal follow-up in primary and specialist respiratory care.

Although the SPPB is not a universally accepted gold standard for frailty, as no such standard exists, it offers several advantages over alternative instruments such as the Fried Frailty Phenotype, the FRAIL scale, and the Clinical Frailty Scale. Its multidomain assessment—covering balance, gait speed, and lower-limb strength—has demonstrated predictive validity for adverse outcomes in COPD and is recommended by ERS/ATS guidelines. These features support its use as the reference measure in our study and provide a practical benchmark for comparing other frailty screening tools.

For clinicians, these results offer practical guidance on which frailty tool to select in different real-world scenarios. Among the three tools, TUG was the most robust proxy for the SPPB, demonstrating the highest agreement, predictive accuracy, and longitudinal stability. Its multidomain assessment—integrating lower-limb strength, gait speed, and balance—closely mirrors the components of the SPPB. Given that gait speed is a key component of both the SPPB and TUG, our findings also underscore the importance of assessing gait as a simple yet informative marker of functional status and frailty risk in patients with COPD.

In clinical practice, this means that the TUG may serve as the preferred option when a comprehensive SPPB assessment is desired, but time or resources are limited.

The 30secSTS showed comparable accuracy, particularly when the clinical aim is to rule out frailty. This makes it an attractive tool in busy outpatient clinics or rehabilitation triage, where rapid exclusion of frailty is often prioritized. HGS, while easy to administer in space-limited settings, lacks assessment of balance and lower-limb function and showed a lower NPV, limiting its utility as a standalone tool. Nevertheless, HGS may retain value as a quick preliminary screen, especially in settings where patient mobility or safety concerns restrict more demanding tests. All three tools are time-efficient, but practical considerations such as available space, patient mobility, and the specific screening objective (identification vs. exclusion) should guide tool selection. Ultimately, our comparative data provide clinicians with evidence-based options, allowing tailoring of frailty assessment to the patient population, clinical setting, and intended use.

Importantly, physical frailty in patients with COPD of any severity should not limit access to rehabilitation but rather prompt early identification and tailored intervention. Future research should focus on optimization of early detection pathways, evaluation of tool performance in clinical settings, and testing of rehabilitation models that effectively address frailty in the respiratory patient population.

This study has several strengths. It is the first to prospectively examine frailty transitions and compare ERS/ATS-recommended screening tools in patients with COPD abstaining from PR over 12 months. Repeated SPPB use as a reference standard and inclusion of multiple frailty instruments enhance internal consistency and clinical relevance. The pragmatic cohort design also mirrors real-world conditions in which these tools are applied. However, some limitations must be acknowledged. The study included only patients with stable but clinically advanced COPD, limiting data generalizability to those with milder disease, recent exacerbations, or cognitive impairment. Matched healthy controls were not included. In the absence of a universally accepted gold standard for physical frailty, the SPPB was used as a reference. Yet the discrepancies observed between instruments raise important questions about their construct validity and the extent to which they capture physical frailty as a distinct biological syndrome characterized by diminished physiologic reserve. Moreover, while the SPPB was originally validated in populations aged ≥65 years, it has been widely applied in chronic disease cohorts, including younger adults. In this context, a potential ceiling effect may limit its sensitivity to detect frailty in younger participants. Still, in our cohort, SPPB-defined pre-frailty was observed equally in those <65 and ≥65 years, suggesting that the tool may capture low physical reserve or poor conditioning in younger individuals—an important clinical phenotype given its potential to increase vulnerability to frailty later in life. In settings where a ceiling effect is suspected, complementary measures with established normative data across younger age ranges (e.g., handgrip strength, 30secSTS) may provide additional insight and help guide preventive strategies.

## 5. Conclusions

Physical frailty was prevalent and persistent among patients with severe COPD who declined PR. The ERS/ATS-recommended screening tools showed fair to moderate agreement with SPPB and demonstrated stable predictive performance over 12 months. TUG was the most robust proxy, though tool selection should be guided by clinical context and purpose. Routine frailty screening may help identify physically frail patients unlikely to improve without tailored exercise intervention.

## Figures and Tables

**Figure 1 jcm-14-06434-f001:**
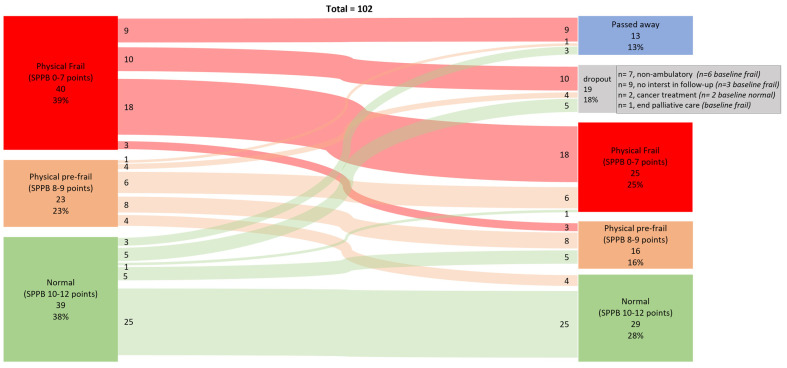
Data are numbers and percentages of participants classified as normal (Short Physical Performance Battery (SPPB) score 10–12 points), physically pre-frail (SPPB score 8–9 points), and physically frail (SPPB score 0–7 poinst) at baseline and 12-month follow-up and classification change.

**Table 1 jcm-14-06434-t001:** Demographics and characteristics at baseline (T0).

Variables	All (n = 102)	Normal SPPB (10–12) (n = 39)	Physical Pre-Frail SPPB (8–9) (n = 23)	Physical Frail SPPB (≤7) (n = 40)
Female sex, n (%)	61 (50)	18 (46)	13 (57) ^a,b^	30 (75) ^a,b^
Age, yr.	70.4 ± 8.7	66.5 ± 8.7	71.1 ± 7.3 ^a^	73.9 ± 8.0 ^a^
Marital status, n (%)				
Married/living with partner	50 (49.0)	22 (56)	12 (52)	16 (40)
Living alone	52 (51.0)	17 (44)	11 (48)	24 (60)
Body mass index, kg/m^2^	25.1 ± 5.9	25.1 ± 5.9	25.8 ± 6.0	24.7 ± 6.1
FEV_1_, % predicted	34.0 ± 11.6	33.8 ± 11.9	31.3 ± 8.6	36.0 ± 12.5
GOLD I/II/III/IV, %	0/2/56/42	0/3/51/46	0/0/67/33	0/2/63/35
A/B/C/D, %	0/57/3/40	0/72/2/26	0/52/0/48	0/45/5/50
LTOT, n (%)	13 (13)	3 (8)	2 (10)	8 (19)
MRC score, median (IQR)	4 (3–5)	4 (3–5)	3 (3–4)	4 (3–5)
Smoking status, %				
Never/Former/Current	3/79/18	3/82/15	9/67/24	0/83/17
Pack-year history, yrs	44 ± 22	43 ± 24	50 ± 26	41 ± 19
Charlson Comorbidity Index score,	2.6 ± 1.3	2.4 ± 1.2	2.9 ± 1.4	2.7 ± 1.3
Walking aid, yes, n (%)	49 (48.0)	7 (18.0)	13 (56.0)	29 (73.0)
Short Physical Performance Battery,				
total score points, median (IQR)	8.0 (5–10)	11.0 (10–12)	9.0 (8–9) ^a,b^	4.5 (3–6) ^a,b^
*Balance, points*	*4.0 (2–4)*	*4.0 (4–4)*	*4.0 (4–4)*	*2.0 (1–3)*
*3 m gait speed, points*	*2.0 (1–4)*	*4.0 (3–4)*	*2.0 (2–3)*	*1.0 (1–2)*
*5-time sit-to-stand, points*	*2.5 (1–4)*	*4.0 (3–4)*	*3.0 (2–3)*	*1.0 (0–1)*
Hangrip strength (dominant), kilo	28.2 ± 10.9	35.5 ± 10.5	27.0 ± 10.3 ^a^	21.8 ± 6.6 ^a^
30sec sit-to-stand, repetition	9.3 ± 6.6	14.6 ± 5.4	9.0 ± 2.4 ^a,b^	4.2 ± 4.2 ^a,b^
Timed-Up-and-Go, seconds, median (IQR)	9.1 (6.6–13)	6.3 (5.6–7.9)	9.5 (7.5–10.9) ^a,b^	13.5 (10.8–22.1) ^a,b^
Steps/Day, median (IQR) (n = 77)	2018 (815–3629)	3230 (2041–4357)	1626 (1194–2895) ^a^	1156 (640–2445) ^a^
CAT, score points	19.2 ± 6.8	17.1 ± 6.5	20.0 ± 6.4	20.9 ± 6.9 ^a^
HADS—Anxiety, score point, median (IQR)	5 (3–8)	4 (3–7)	6 (3–10)	5 (2–7)
HADS—Depression, score point, median (IQR)	3 (1–6)	2 (1–5)	4 (1–7)	3 (2–7)
EQ-5D, VAS score	57.2 ± 19.1	65.6 ± 18.3	53.0 ± 16.6 ^a^	51.6 ± 18.8 ^a^
EQ-5D, index score	0.7 ± 0.2	0.7 ± 0.2	0.7 ± 0.2	0.6 ± 0.2 ^a^
Multiple Fatigue Inventory, total score	58.7 ± 14.9	51.4 ± 13.4	62.0 ± 14.0 ^a^	63.9 ± 14.2 ^a^
Brief Pain Inventory				
Pain severity score, median (IQR)	2.3 (0.0–3.6)	2.0 (0.5–3.3)	2.1 (0.0–3.3)	2.8 (0.1–4.8)
Pain Inference score, median (IQR)	1.4 (0.0–3.6)	1.5 (0.1–2.9)	2.2 (0.0–4.6)	2.4 (0.0–3.9)
Pulmonary rehabilitation, naïve/previously, %	38/62m	49/51	36/65	30/70

Data are presented as mean ± SD, except where otherwise indicated. ^a^
*p* < 0.05 between the normal SPPB and marked group, and ^b^
*p* < 0.05 between the pre-frail and frail groups. Abbreviations: FEV_1_% predicted, forced expiratory volume in the first second; GOLD, Global Initiative for Chronic Obstructive Lung Disease; LTOT, long-term oxygen therapy; MRC, Medical Research Council; CAT, COPD Assessment Test; HADS, Hospital Anxiety and Depression Scale; EQ-5D, Euro-Qol 5-dimension.

**Table 2 jcm-14-06434-t002:** Change in test score from baseline (T0) to 12-month follow-up (T1) †.

Variables	All (n = 70)	Normal SPPB (10–12) (n = 31)	Physical Pre-Frail SPPB (8–9) (n = 18)	Physical Frail SPPB (≤7) (n = 21)
SPPB total score, median (IQR)	0.0 (−1.0; 0.0)	0.0 (−1.0; 0.0)	0.0 (−2.3; 1.3)	0.0 (−1.0; 0.0)
*Balance, points*	*0.0 (0.0; 0.0)*	*0.0 (0.0; 0.0)*	*0.0 (−0.3; 0.0)*	*0.0 (−0.5; 0.0)*
*3MGS, points*	*0.0 (−0.25; 0.0)*	*0.0 (−0.25; 0.0)*	*0.0 (−1.0; 0.3)*	*0.0 (0.0; 0.5)*
*5STS, points*	*0.0 (−1.0; 0.0)*	*0.0 (−1.0; 0.0)*	*0.0 (−2.0; 0.5)*	*0.0 (−0.5; 0.0)*
HGS, kilo, mean (95%CI)	−0.6 (−1.5; 0.3)	−0.5 (−1.7; 0.6)	−0.3 (−1.9; 1.2)	−0.9 (−3.1; 1.3)
30secSTS, repetition mean (95%CI)	−1.4 (−2.3; −0.4) *	−1.3 (−2.8; 0.3)	−1.3 (−3.5; 0.9)	−1.5 (−2.9; −0.1) *
TUG, seconds, median (IQR)	0.2 (−0.4; 1.4) *	0.1 (−0.4; 1.0)	0.4 (−1.1; 1.5)	0.9 (−0.1; 2.8)

† Complete data observations used for the comparisons (n = 70). Data are presented as the difference in median (25th to 75th IQR) or mean (SE 95% CI), as indicated. Differences are presented as (T1-T0). Any statistically significant difference between T1 and T0 denoted * *p* < 0.05. Abbreviations: SPPB, Short Physical Performance Battery; 3MGS, 3 m gait speed; 5STS, 5 time sit-to-stand time; HGS, hand grip strength; 30secSTS, 30 s sit-to-stand test; TUG, Timed-Up-and-Go.

**Table 3 jcm-14-06434-t003:** Predictive value of ers/ats-recommended frailty screening tests compared to sppb classification at baseline (t0) and 12-month follow-up (t1).

Test	PPV Baseline	NPV Baseline	PPV 12-Month †	NPV 12-Month †
HGS (>1 SD below normal)	0.77 **	0.49 *	0.78 **	0.42 ***
30secSTS (>1 SD below normal)	0.75 **	0.73 **	0.82 *	0.81 *
TUG (>8 s)	0.85 *	0.71 **	0.84 *	0.74 **

† Complete data observations were used for the comparisons (n = 70). * High > 0.80, ** Moderate 0.60–0.79, *** Low agreement < 0.60. Abbreviations: PPV, positive predictive value; NPV, negative predictive value; HGS, hand grip strength; 30secSTS, 30 s sit-to-stand test; TUG, Timed-Up-and-Go.

## Data Availability

The data presented in this study are available on request from the corresponding author.

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
