# Peer review of "Prevalence, Persistence, and Agreement of Physical Frailty Tools in Patients with Severe COPD Declining Pulmonary Rehabilitation: An Exploratory 1-Year Prospective Cohort Study†"

_jcm, 2025, doi:10.3390/jcm14186434_

Round 1
Reviewer 1 Report
Comments and Suggestions for Authors
I appreciate the opportunity to review this article. Overall, I find it relevant and well written. However, there are some points that should be addressed:
- In fact, the mean FEV₁ value of 33% indicates that this is a very severe group of patients. I believe this should be clearly reflected in both the title and the main text. Most of the existing literature includes patients with varying degrees of COPD, and this study represents a valuable opportunity to present evidence specifically in this severely affected population.
- In Figure 1, the abbreviation “SPPS” appears, which is incorrect. It should be corrected to “SPPB” to match the terminology used throughout the manuscript.
- Gait speed is incorporated in several functional assessments of patients with COPD. I suggest the authors add a sentence in the manuscript discussing their findings in relation to gait speed, as it may provide additional clinical context and relevance.
- The authors state that the SPPB is not the gold standard for frailty assessment. I suggest they expand on this point by presenting other available frailty assessment tools, perhaps detailing the advantages of using the SPPB over alternative instruments. This would strengthen the rationale for its use as the reference measure in this study.
Author Response
Dear Reviewer, Section Managing Editor Miana Zhou
and Special Issue Editor Prof. Dr. Akira Tamaki,
August 25 2025
We thank the editor for the opportunity to submit a revised version of our manuscript entitled “Prevalence, Stability, and Agreement of Physical Frailty Tools in Patients with COPD Declining Pulmonary Rehabilitation: An exploratory 1-Year Prospective Cohort Study”. We thank the reviewers for thorough and constructive comments that have helped us clarify and improve our manuscript. All changes made in the manuscript are marked in the track changes manuscript. Please find our point-by-point response to specific reviewer comments below (responses in italic bold).
Dear Professor Hansen,
I hope this email finds you well. To process your manuscript smoothly, could you please confirm the following information?
--
1. Informed Consent for Participation
We have noticed that your research contains studies involving human participants. To proceed further with your manuscript, please provide us with a blank version of the consent form that your participant(s), (or their guardian(s)), signed indicating their agreement to take part in the experiment.
Please note that the form cannot contain identifying information, so it needs to be blank. If the informed consent form is not in English, please provide the translated version. The document is for our records and will not be made publicly available.
We confirm that all participants, as required by law and ethics, signed the standardized consent form provided by the Danish National Committee on Health Research Ethics. A blank English translation of the form (“S1_Consentform_english”) is submitted separately with the revised manuscript and reviewer response letter.
- Open review
We notice that you chose "Open Review" when submitting your paper.
Please note that "Open Review" is different from "Open Access".
"Open Review" means that the review reports and the responses to reviewers will be published online together with the paper if the paper is accepted for publication. In other words, if you select "Open Review", readers can see your published paper as well as the review reports and your responses. Please confirm whether you would like to select "Open Review" for your paper or not.
- Figure
There seems lack Figure 2 in the manuscript, please check and add it.
Thank you for bringing this to attention. This is a typo error, thus there is no figure 2. We have corrected the typo error in the manuscript. Please read correction on p.6 line252 in the track changes manuscript.
---
We look forward to hearing from you.
Kind regards,
Miana Zhou
Section Managing Editor
Reviewer 1
Comments and Suggestions for Authors
I appreciate the opportunity to review this article. Overall, I find it relevant and well written. However, there are some points that should be addressed:
- In fact, the mean FEV₁ value of 33% indicates that this is a very severe group of patients. I believe this should be clearly reflected in both the title and the main text. Most of the existing literature includes patients with varying degrees of COPD, and this study represents a valuable opportunity to present evidence specifically in this severely affected population.
We agree that the severity can be emphasized and clarified for the reader.
We have modified the title, that now reads: “Prevalence, Stability, and Agreement of Physical Frailty Tools in Patients with Severe COPD Declining Pulmonary Rehabilitation: An exploratory 1-Year Prospective Cohort Study”
Furthermore, we have emphasized “severe COPD” throughout the manuscript and in the conclusion for both abstract and manuscript. We hope that the corrections are clear and addresses your observation sufficiently. Please read track change specification on severity throughout the manuscript.
- In Figure 1, the abbreviation “SPPS” appears, which is incorrect. It should be corrected to “SPPB” to match the terminology used throughout the manuscript.
Thank you for this observation. The typo error has been corrected from SPPS to SPPB. See correction in track changes manuscript p. 9, line: 324-325.
- Gait speed is incorporated in several functional assessments of patients with COPD. I suggest the authors add a sentence in the manuscript discussing their findings in relation to gait speed, as it may provide additional clinical context and relevance.
We thank the reviewer for this insightful suggestion. While detailed analysis of gait speed was beyond the primary scope of our study, we agree that it provides valuable clinical context. We have therefore added a brief sentence in the Discussion highlighting the relationship between our findings and gait speed, emphasizing its role in functional assessments and frailty screening in patients with COPD. Please read track change manuscript to read the refinement. (p. 9, line: 350-354)
- The authors state that the SPPB is not the gold standard for frailty assessment. I suggest they expand on this point by presenting other available frailty assessment tools, perhaps detailing the advantages of using the SPPB over alternative instruments. This would strengthen the rationale for its use as the reference measure in this study.
We have expanded the Discussion to include a brief overview of alternative frailty assessment tools and clarified the rationale for using the SPPB as the reference measure in our study, highlighting its multidomain assessment, predictive validity in COPD, and recommendation in the ERS/ATS guidelines. Please read track change manuscript to read the refinement. (p. 9, line: 339-345)
Reviewer 2
Comments and Suggestions for Authors
General
The present study investigated the prevalence and changes in physical frailty among patients with COPD who declined pulmonary rehabilitation. The study also aimed to determine the association between frailty and 12-month all-cause hospitalizations and mortality. Additionally, it sought to evaluate the agreement and predictive value of various frailty screening tests, including the Short Physical Performance Battery, handgrip strength, 30-second sit-to-stand, and Timed-Up-and-Go. The study is well-designed, and its connection to a larger project—which compared patients with advanced COPD who either opted into or out of pulmonary rehabilitation—provides valuable context. However, I have a few considerations that need to be addressed.
Title
I would like to suggest a minor change to the title. In a scientific context, "stability" can imply a fixed, unchanging state. "Persistence," on the other hand, is the more precise term for a characteristic that continues over a period of time. As your study clearly demonstrates that frailty persisted in the majority of patients, using "persistence" in the title would make the study's main message more accurate and scientifically robust.
We acknowledge your point and suggestion for change. The word Stability has been replaced with the word Persistence in the title. Please read the corrected title in the track change manuscript (p. 1 line 2).
Introduction section
The introduction provides a comprehensive overview; however, for clarity and impact, I suggest a more streamlined approach. A more direct articulation of the current knowledge gap and the study's specific importance would better highlight the manuscript's contribution to the field.
We thank the reviewer for this constructive suggestion. We have made more clearly articulating regarding the knowledge gap and the study’s specific contribution. To address this, we have slightly streamlined the final paragraph of the Introduction to more directly emphasize the limited evidence regarding frailty stability and the agreement of ERS/ATS-recommended screening tools in patients with COPD who decline pulmonary rehabilitation. We believe this adjustment improves clarity and impact while preserving the comprehensive context provided in the preceding paragraphs.
Please read trach changes (p. 2, lines 86-91).
Please review the bibliographic references. For instance, the definition of frailty is attributed to a specific article. However, upon closer inspection, it appears this article cites two other primary sources for that same definition. To maintain academic rigor, the original articles that introduced this definition must be cited directly.
We thank the reviewer for this important point. To ensure proper attribution, we have now cited the original article by “Fried LP, Tangen CM, Walston J, et al., Frailty in Older Adults: Evidence for a Phenotype (J Gerontol A Biol Sci Med Sci. 2001;56(3):M146–M157)”, which as one of the first operationalized the physical frailty phenotype. Figures as reference 3 in the mnuscript.
We note that Fried et al. also co-authored one of the previously cited sources; however, citing the 2001 article directly ensures accurate attribution to the original definition.
The phrase 'ERS/ATS-recommended physical frailty screening tests' is mentioned throughout the manuscript. To ensure proper attribution and to allow readers to find the original source, please include the specific citation for the ERS/ATS guidelines or consensus document that established these tests.
We thank the reviewer for this suggestion. We have now inserted the specific citation for the ERS/ATS guidelines the first time the phrase “ERS/ATS-recommended physical frailty screening tests” appears, at the end of the Introduction section. Please find the inserted reference in the track change manuscript p.2, line 90.
Methods section
The ethical statement on lines 105-106 is an important but limited summary. For enhanced clarity and to provide readers with a more complete understanding of the ethical framework, it is suggested that the current statement be replaced with the more detailed information found on lines 364-366.
The revised text would read as follows:
"The Danish Data Protection Agency approved the research database (P-2019-730). As this study was noninterventional, ethical approval was not required according to Danish law. Informed verbal and written consent were obtained from all subjects involved in the study."
This change would consolidate essential details about data protection and informed consent in a single, prominent location.
Thank you for bringing this to attention. We have inserted the more detailed text into the manuscript. Please read trach changes (p. 3, lines 110-114).
The finding that patients classified as pre-frail and frail were predominantly older is notable. Given that the SPPB cut-off scores were originally validated in a population aged 65 and older, it is plausible that these thresholds may not possess adequate sensitivity to identify frailty in younger participants. It would be valuable for the authors to discuss this as a potential methodological limitation, commenting on how the application of these cut-offs might have influenced the classification of individuals under the age of 65.
We thank the reviewer for this fair point. We agree that the SPPB was originally validated in populations aged ≥65 years, which may limit its sensitivity to identify frailty in younger adults. At the same time, the SPPB has been applied across a range of chronic disease populations, including individuals <65 years. In younger adults, the SPPB may be subject to a ceiling effect, thereby reducing its ability to discriminate early frailty. Yet, in our cohort, the prevalence of pre-frailty was similar in those <65 and ≥65 years (21%), whereas frailty was markedly less frequent in younger patients (14% vs. 52%). This pattern suggests that SPPB-defined pre-frailty in younger adults (<65 years) may capture low physical reserve or poor conditioning—potentially representing “real” pre-frailty. From a clinical perspective, this phenotype remains important, as it may indicate increased vulnerability to developing frailty later in life. In cases where a ceiling effect is suspected but low physical reserve or poor conditioning is present, tools with normative data extending to younger age ranges (down to 25 years in Danish reference values), such as handgrip strength and the 30-secSTS, may provide complementary insight and help guide early preventive initiatives. We have now expanded the Discussion to include this consideration in the strengths and limitations section (p. 10, line 383-392).
The EQ-5D VAS is included in the results, but its use is not described in the methods section. Given the importance of assessing quality of life in this condition, it is crucial that the authors provide a description of this instrument and a rationale for its inclusion, or at least a specific citation, in the methods section of the manuscript.
We thank the reviewer for this important observation. The EQ-5D VAS and the other PROMs (HADS, BPI, MFI-20) were prespecified to characterize the cohort and provide clinical context rather than to serve as outcomes of interest in this exploratory study. To maintain focus on the head-to-head comparison of frailty tools and to remain within the journal’s word limit, we have kept the PROMs description concise and limited to brief instrument identification. We hope our intentions are now clear, corrections acceptable and addresses the point raised satisfactory. See revised subsection “descriptive variables” , in track changes (p.4, lines 185–193).
Results section
The results section is well-organized and presented concisely. The figures and tables significantly facilitate the comprehension of the study's findings.
For enhanced clarity, it is suggested that the labels for the baseline and 12-month time points be integrated directly into Figure 1, rather than being confined to the figure legend. This would make the figure more self-explanatory for the reader.
We completely agree with your suggestion. This was already integrated into the figure 1.
By mistakes the heading was not included when converting the figure into a jpeg picture.
Thanks to your observation the headings are now include and the figure and is more self-explanatory.
Discussion section
Line 312-328: These lines are the most critical part of the discussion, especially for a clinical audience. It provides a direct and practical comparison of frailty screening tools, which is of great value for day-to-day use. I suggest revising the text to further highlight the clinical relevance of these findings. Emphasizing the head-to-head comparison and the specific utility of each test could be made even more explicit and digestible for the clinical reader.
We thank the reviewer for this important and constructive comment. We fully agree that this section represents a key opportunity to enhance the clinical relevance of our findings. In response, we have retained the original structure of this section but added new bridging text to explicitly highlight the practical implications for clinical decision-making. Specifically, we now emphasize how each tool (TUG, 30secSTS, and HGS) may be applied in real-world respiratory care, including their relative strengths and limitations in different clinical contexts (see revised Discussion, p. 9-10 line 354-366). These additions aim to make the head-to-head comparisons more digestible and actionable for clinicians.

Reviewer 2 Report
Comments and Suggestions for Authors
General
The present study investigated the prevalence and changes in physical frailty among patients with COPD who declined pulmonary rehabilitation. The study also aimed to determine the association between frailty and 12-month all-cause hospitalizations and mortality. Additionally, it sought to evaluate the agreement and predictive value of various frailty screening tests, including the Short Physical Performance Battery, handgrip strength, 30-second sit-to-stand, and Timed-Up-and-Go. The study is well-designed, and its connection to a larger project—which compared patients with advanced COPD who either opted into or out of pulmonary rehabilitation—provides valuable context. However, I have a few considerations that need to be addressed.
Title
I would like to suggest a minor change to the title. In a scientific context, "stability" can imply a fixed, unchanging state. "Persistence," on the other hand, is the more precise term for a characteristic that continues over a period of time. As your study clearly demonstrates that frailty persisted in the majority of patients, using "persistence" in the title would make the study's main message more accurate and scientifically robust.
Introduction section
The introduction provides a comprehensive overview; however, for clarity and impact, I suggest a more streamlined approach. A more direct articulation of the current knowledge gap and the study's specific importance would better highlight the manuscript's contribution to the field.
Please review the bibliographic references. For instance, the definition of frailty is attributed to a specific article. However, upon closer inspection, it appears this article cites two other primary sources for that same definition. To maintain academic rigor, the original articles that introduced this definition must be cited directly.
The phrase 'ERS/ATS-recommended physical frailty screening tests' is mentioned throughout the manuscript. To ensure proper attribution and to allow readers to find the original source, please include the specific citation for the ERS/ATS guidelines or consensus document that established these tests.
Methods section
The ethical statement on lines 105-106 is an important but limited summary. For enhanced clarity and to provide readers with a more complete understanding of the ethical framework, it is suggested that the current statement be replaced with the more detailed information found on lines 364-366.
The revised text would read as follows:
"The Danish Data Protection Agency approved the research database (P-2019-730). As this study was noninterventional, ethical approval was not required according to Danish law. Informed verbal and written consent were obtained from all subjects involved in the study."
This change would consolidate essential details about data protection and informed consent in a single, prominent location.
The finding that patients classified as pre-frail and frail were predominantly older is notable. Given that the SPPB cut-off scores were originally validated in a population aged 65 and older, it is plausible that these thresholds may not possess adequate sensitivity to identify frailty in younger participants. It would be valuable for the authors to discuss this as a potential methodological limitation, commenting on how the application of these cut-offs might have influenced the classification of individuals under the age of 65.
The EQ-5D VAS is included in the results, but its use is not described in the methods section. Given the importance of assessing quality of life in this condition, it is crucial that the authors provide a description of this instrument and a rationale for its inclusion, or at least a specific citation, in the methods section of the manuscript.
Results section
The results section is well-organized and presented concisely. The figures and tables significantly facilitate the comprehension of the study's findings.
For enhanced clarity, it is suggested that the labels for the baseline and 12-month time points be integrated directly into Figure 1, rather than being confined to the figure legend. This would make the figure more self-explanatory for the reader.
Discussion section
Line 312-328: These lines are the most critical part of the discussion, especially for a clinical audience. It provides a direct and practical comparison of frailty screening tools, which is of great value for day-to-day use. I suggest revising the text to further highlight the clinical relevance of these findings. Emphasizing the head-to-head comparison and the specific utility of each test could be made even more explicit and digestible for the clinical reader.
Author Response

(The authors gave the same response as above.)

Round 2
Reviewer 2 Report
Comments and Suggestions for Authors
Dear Author's,
Thank you for sending the revised version of your manuscript, "Prevalence, Stability Persistence, and Agreement of Physical Frailty Tools in Patients with Severe COPD Declining Pulmonary Rehabilitation: An exploratory 1-Year Prospective Cohort Study".
I have carefully reviewed the new version and your response to the reviewers' comments. I am pleased to confirm that you have addressed all my previous suggestions satisfactorily. The changes have improved the manuscript's clarity and quality.
I have no further comments at this time.